# Does Attentional Style Moderate the Relationship between Time Perspective and Social Network Addiction? A Cross-Sectional Study on a Sample of Social Networking Sites Users

**DOI:** 10.3390/jcm10173983

**Published:** 2021-09-02

**Authors:** Silvana Miceli, Fabrizio Scrima, Maurizio Cardaci, Giuseppe Quatrosi, Luigi Vetri, Michele Roccella, Barbara Caci

**Affiliations:** 1Department of Psychology, Educational Science and Human Movement, University of Palermo, 90133 Palermo, Italy; silvana.miceli56@unipa.it (S.M.); maurizio.cardaci@unipa.it (M.C.); michele.roccella@unipa.it (M.R.); 2Département de Psychologie, Universitè de Rouen, 76130 Mont-Saint-Aignan, France; fabrizio.scrima@univ-rouen.fr; 3Department of Health Promotion, Mother and Child Care, Internal Medicine and Medical Specialties (ProMISE), University of Palermo, 90127 Palermo, Italy; giuseppe.quatrosi01@community.unipa.it; 4Oasi Research Institute-IRCCS, 94018 Troina, Italy; luigi.vetri@oasi.en.it

**Keywords:** attention, attentional style, time perspective, social network addiction, time orientations, psychology, psychiatry, social media, Internet, behavioral addiction

## Abstract

The present study investigates the role of attentional style as a moderator variable between temporal perspective and social network addiction, since little is known about users’ cognitive variables involved in this kind of addictive behavior. To achieve this goal, a sample of 186 volunteers and anonymous social networking sites users (M = 34%; F = 66%; *M*_age_ = 22.54 years; SD = 3.94; range: 18 ÷ 45 years) participated in a cross-sectional study. All participants filled out self-report instruments measuring temporal perspective, internal vs. external attentional style, and social network addiction. The results align with the previous literature and show that present fatalistic and past negative time orientations are associated with social network addiction, whereas the future is a negative precursor. Moreover, a four-step hierarchical regression analysis showed that internal attentional style is a significant moderator of the relationship between high levels of temporal perspective and a high level of social network addiction. This result suggests that social network-addicted users are oriented toward internal stimuli such as their intrusive thoughts or feelings and that social network addiction is similar to obsessive compulsive disorders, depression, or anxiety. Despite its limitations, the present study could contribute to the efforts of clinicians, psychiatrists, psychologists, teachers, and all those who seek to combat social network addiction in developing treatment programs to reduce its harmful effects.

## 1. Introduction

The use of the Internet and its social networking sites (SNSs), such as Facebook, Twitter, WhatsApp, and Instagram, is a worldwide phenomenon that involves more than 4 billion people in the world [1]. Due to this high prevalence, the daily use of this kind of social application represents a sort of “normal” behavior nowadays [2,3], but the excessive use of these technological platforms might also become an “uncontrolled” behavior leading to a form of compulsive addiction. The literature has provided different terminologies to define this compulsive use, such as Internet addiction [4], pathological Internet use [5], problematic Internet use [6], and social network addiction [7,8,9], but all of them fit the behavioral addiction model framework [10]. In the current research, we considered the terms mentioned above as being quite similar. However, we adopted the definition of social network addiction provided by Andreassen and Pallesen [11] (p. 4054): “being overly concerned about social networking sites to be driven by a strong motivation to log on to or use Social Networking Sites, and to devote so much time and effort to Social Networking Sites that it impairs other social activities, studies/job, interpersonal relationships, and/or psychological health and well-being.” Social network addiction includes a decline in work and academic productivity [12,13], relational problems [14], and a variety of symptoms of distress [15,16]. The core symptoms of social network addiction are (a) salience—i.e., spending too much time thinking about social networking sites; (b) tolerance—i.e., the urgent need for more and more use; (c) mood modifications—i.e., using social media to reduce feelings of guilt, anxiety, restlessness, and depression, and (d) withdrawal—i.e., feeling distracted, restless, disturbed, irritable and uncomfortable if the use of social networking sites is prohibited. Recent studies [17,18,19] linked social network addiction to time perspectives, considering the well-known association between time perspectives and several types of addiction such as gambling [20], alcohol, drugs, tobacco, and cannabis abuse [21,22].

Time perspective is defined as “the often-non-conscious process whereby the continuous flows of personal and social experiences are assigned to temporal categories, or time frames, that help to give order, coherence, and meaning to those events” [23] (p. 1271). Following the time orientations suggested by Zimbardo and Boyd [23]—i.e., present-fatalistic, present-hedonistic, past-negative, past-positive, and future—there is increasing evidence that present-fatalistic and past-negative could be the most significant positive predictors for social network addiction, in line with the prior literature [17,18,24]. Since little is known about the cognitive variables related to social network addiction, the present study investigates the role of attentional style as moderator variables between time perspectives and social network addiction. As stated by prior literature, attentional style is related both with time perspectives [23] and with addictive behaviors such as social network addiction [25]; therefore, we retain that attentional style might be responsible for individual variations in the relationship between time perspectives (all dimensions) and social network addiction. More specifically, internal vs. external attentional style might be responsible for the ways in which people select and focus their attention on their internal thoughts about past, present, or future life events and external interactive stimuli provided by social networking sites. This process, in turn, might lead people to also become more or less addicted. Thus, we seek to verify whether individual differences in attentional style, interacting with time perspectives, and leading people to focus and select both internal or external information about their past, present, or future actions and experiences moderate individuals’ social network addiction levels.

## 2. Theoretical Background

### 2.1. Time Perspective and Social Network Addiction

According to Zimbardo and Boyd [23], time perspective has both positive and negative aspects depending on the way people focus on their experiences from the past, present, and future. Time perspective influences individuals’ actions and decisions unconsciously, leading them to to develop a stable preference for one or more time perspective under their experiences of success/failure, happiness, stressful life situations, family, or cultural backgrounds. Past-negative-oriented individuals continually relieve their negative past experiences, avoid changes, and keep alive the memory of their failures. They have few close friends, often feel depression, anxiety, and have low self-esteem [23]. Past-negative individuals are more likely to be more alcohol and drug-addicted than people with a different time orientation [26]. People temporally framed as past-positive are more focused on memories of events that evoke pleasant experiences and use such memories to feel good in the present. Present-fatalistic people believe that they cannot change the present and are guided passively by external events that they experience ineluctably. Contrarily, those who are present-hedonistic prioritize pleasure and immediate gratification. They actively carry out stimulating activities and do not reflect upon future consequences. Both PF and present-hedonistic individuals tend to participate in at-risk behaviors related to sexual habits [27], driving [28], alcohol and drug misuse [21], suicide [29] and chronic homelessness [30]. The present-fatalistic orientation is also associated with aggression, anxiety, depression [23], and avoidant procrastination, while present-hedonistic is associated with arousal procrastination [31]. Finally, people focused on the future are continually striving to plan and achieve future goals, but sometimes they cannot fully live the present [23]. Overall, future time orientation has been associated with less problematic behaviors [32], even if people who emphasize their future goals might lose spontaneity do not enjoy the present [33].

Recent studies have linked time perspective to various forms of behavioral addiction, such as pathological Internet use (PIU) [17], internet Addiction (IA) [18], and online gaming addiction [19]. More specifically, the study of Chittaro and Vianello [17] found associations between PIU and past-negative (*r* = 0.37, *p* < 0.001) or present-fatalistic (*r* = 0.26, *p* < 0.01) orientations in a sample of young Italian Facebook users (*n* = 149). Moreover, the past-negative and present-fatalistic orientations have emerged as predictors of PIU: participants (especially women) scoring higher in both scales obtained higher values on the mood regulation subscale. Thus, the authors suggested that people with a negative time perspective both about the past and the present use the Internet to alleviate their negative moods. Furthermore, a moderate correlation between past-negative orientation and the preference for the online social interaction subscale has suggested that past-negative people are addicted to the Internet, since it is a safer and better place for social interactions. Similar results were reported by Przepiorka and Blachnio [18] in a study carried out on a sample of Polish Internet users (*n* = 756). In addition, the authors found positive and robust associations between past-negative or present-fatalistic perspectives and IA or FA. More recently, Kim, Hong, Lee, and Hyun [34] have reported that the present-hedonistic and present-fatalistic perspectives are significantly positive predictors of IA. Similarly, Lukavska [19] corroborated the predictive role of a past-negative perspective on Internet gaming disorder (IGD), recently included in the “Emerging Measures and Models” section of the latest version of the Diagnostic Statistical Manual of Mental Disorder (DSM-5, American Psychiatric Association [APA], 2013) [7]. On the contrary, Future was not correlated with PIU [17] and was negatively associated with Internet and Facebook Addictions [18].

### 2.2. Time Perspective and Attention

We might define time perspective in cognitive terms as a mental ability to switch effectively between time horizons [23]. Consequently, scholars have suggested that time perspective’s flexible and effective use may depend on cognitive resources such as attention. Individual differences in top-down or bottom-up attentional style might influence time perspective [23]. Top-down attentional style corresponds to goal-directed, controlled attention processes, while bottom-up attentional style is the natural attraction of attention towards novel, salient, and unexpected stimuli [35]. People with a top-down approach might transcend compelling stimulus forces in the immediate living space and delay apparent gratification sources that might lead to undesirable consequences, orienting more toward reconstructing the past or constructing the future. In contrast, those who tend to be primarily bottom-up might be more influenced by the sensory, biological, and social qualities associated with the salient elements of the immediate environment, and so orienting more toward their present time [23]. The top-down and bottom-up attentional styles can both be externally or internally oriented [36]. Externally oriented top-down attention drives individuals to intentional, goal-directed processing of the external environmental stimuli, while internally oriented top-down attention is involved in intentional and goal-directed internal stimuli [37,38]. The bottom-up external attentional style involves the sudden and unexpected attention capture of stimuli in the immediate environment. The bottom-up internal attentional style catches possible intrusive thoughts that are unintended, that often interfere with ongoing activity, and that are difficult to control [39]. Internal versus external attentional style and top-down versus bottom-up attentional processes are undoubtedly associated and can interact differently. For example, while performing a visual detection task (external top-down attention), a person can be distracted by a sudden noise (external bottom-up attention) or an intrusive thought (internal bottom-up attention).

Similarly, when intentionally remembering the past or planning the future (internal top-down attention), we can be distracted by a sudden noise or an intrusive thought. Indeed, in the top-down or bottom-up processing of internal/external stimuli, the functions of alerting, orienting, and executive control of attention work together [40]. Despite these theorized relations between time perspective and attentional style [23], few empirical studies there are in the existing literature on this theme. Prior works have only reported that the past-negative and present-fatalistic perspectives are negatively related to cognitive inhibition, which is a part of executive control [41,42]. More specifically, the authors suggested that negative scores on cognitive inhibition might depend on the significant connection between past-negative and present-fatalistic perspectives with negative emotionality, especially neuroticism and anxiety [23,42].

### 2.3. Attention and Social Network Addiction

A recent study by Nikolaidou, Fraser, and Hinvest [25] reported that attention could affect social network addiction, too. Indeed, authors showed that high social network addiction individuals focus more on processing visual stimuli related to social networking sites (e.g., Facebook, Instagram, and WhatsApp logos) than control images (e.g., everyday objects such as a cup, clock, and umbrella). Furthermore, this cognitive attentional bias is higher when people urge to be online, explaining salience symptoms related to raising social networking sites activity as the prevailing thought and preoccupation throughout the day. Salience could lead to craving to go online, and consistent with addiction theories, it could initiate a vicious circle between craving and attention [43,44]. Other studies suggested that mood modification, another core symptom of social network addiction, could result in arousal changes that can lead to increased dopaminergic activity that can further enhance attention toward social networking site-related cues and the urge to go online [45]. From this point of view, social network addiction is quite similar to impulse control disorders [46], and prior studies have shown that attention deficit disorder is the most common psychiatric disorder among adolescents with IA [47,48,49].

## 3. Research Goals and Hypotheses

Consistently with the above-mentioned cross-sectional studies [17,18,19,34], the present study aims to test a moderation model, in which internal or external attentional styles moderate the association between each of the five dimensions of time perspective and SNA.

The present study assumes that predictors, dependent variables, and the potential moderators would have significant bivariate associations to establish the precondition of moderation. Therefore, we expect that the present-fatalistic, past-positive, past-negative orientations would be positively related to social network addiction, whereas the present-hedonistic and future orientations would be negatively related to it.

Furthermore, we also assume that internal or external attentional styles would be related to each dimension of time perspective and social network addiction.

Figure 1 reports the hypothesized model.

## 4. Materials and Methods

### 4.1. Participants and Procedure

Participants were recruited at the researchers’ university for three weeks during the class hours of their psychology classes. The sample consisted of 186 volunteers and anonymous participants (34% men and 66% women) who were not compensated financially or through additional university credits. All participants completed the assessment procedure of the study in the classroom with an average time of about 45 min. The mean age of the sample was 22.54 years (SD = 3.94; range from 18 to 45 years). About 62% were high school graduates, and 38% were university graduates (first bachelor’s degree). Most participants used Facebook (66.6%), whereas only 29.6% used Twitter, and 3.8% used Instagram, thus meeting our inclusion criterion of being social networking site users. This study was compliant with the Ethical Principles for Conducting Research with Human Participants (Declaration of Helsinki) and the Italian Data Protection Authority. Furthermore, according to the research goals and considering that no data related to health or medical issue were gathered, data collection procedures did not need to be approved by the Human Research Ethics Committee of the research institution. Nevertheless, all participants gave written consent after reading a study information sheet and consent form about the anonymity of data handling.

### 4.2. Instruments

Participants completed a paper and pencil questionnaire consisting of demographic questions (i.e., gender, age, education) and three self-report instruments measuring time perspective, internal vs. external attentional style, and social network addiction.

#### 4.2.1. The Zimbardo Time Perspective Inventory

The Zimbardo Time Perspective Inventory (ZTPI) [23] measured time perspective. It consists of 56 items, in five different subscales (one for each time perspective dimension), with a 5-point Likert response scale (from 1 = Very uncharacteristic to 5 = Very characteristic). We computed the total score for each subscale by averaging the scores obtained by participants for each of the scale items. Thus, high scores correspond to a high level in that time perspective dimension. In the present study, the standardized Cronbach’s alpha coefficients of ZTPI were: 0.83 for past-negative; 0.72 for past-positive; 0.70 for present-fatalistic; 0.77 for present-hedonistic; and 0.70 for future, respectively.

#### 4.2.2. The Attentional Style Questionnaire

The Attentional Style Questionnaire (ASQ) [50] was used to measure attentional style. ASQ is a self-report scale consisting of 12 items with a 5-points Likert response scale (from 1 = Totally disagree to 5 = Totally agree). ASQ measures two dimensions consisting of 6 items each: internal attentional style (example of item: In general, I stay in control of my thoughts and do not let myself get distracted by interfering thoughts) and external attentional style (example of item: I am often the first one to notice something has changed in a room). For each dimension, we computed the total score by averaging the scores obtained by participants for each of the items of the two dimensions. High scores correspond to a high level in the Internal vs. external attentional dimension. In the present study, standardized Cronbach’s alpha coefficients of internal and external attentional style were: 0.71 and 0.80, respectively.

#### 4.2.3. The Social Network Addiction—Italian Scale (SNA-IS)

The Social Network Addiction—Italian Scale was developed based on the Facebook Addiction Italian Questionnaire by Caci et al. [51]. To obtain a broader measure related to the use of various SNSs, we adapted the original instrument which consisted of replacing the term Facebook with the term “Social Network Sites (SNSs).” The scale consists of 16 items with a 5-point Likert response scale (from 1 = totally disagree to 5 = totally agree). This questionnaire offers both an social network addiction total score and four different scores related to specific symptoms, measured in four subscales (four items each): interpersonal irritability (example of item: You try to cut down the amount of time you spend on SNSs and fail); elapsed time (example of item: You find that you stay online on SNSs longer than you intended); social performance impairment (example of item: You neglect household chores to spend more time on the SNSs); and social network anxiety (example of item: You feel depressed, moody, or nervous when you are offline, which goes away once you are back on the SNSs). The total score was computed by averaging the scores obtained by participants for each of the scale items. For each subscale, we computed a score by averaging the scores obtained by participants for each of the subscale items. High total scores correspond to a high level of social network addiction and its specific symptoms. In the present study, exploratory factor analysis was performed, confirming the correlated four-factor solution of the original model (Table 1). Standardized Cronbach’s alpha coefficients were: 0.88 for social network addiction total score, 0.70 for interpersonal irritability, 0.77 for elapsed time; 0.76 for social performance impairment; and 0.76 for social network anxiety, respectively.

### 4.3. Data Analysis

First, data were checked for outliers and missing data, and the internal consistency for all the TP, attentional style, and SNA dimensions were analyzed using Cronbach’s alpha. Then, a four-step hierarchical regression analysis was used to test all the hypotheses. In the first step, the demographic variables (i.e., gender and age) were inserted as covariates; in the second step, we included the five dimensions of TP; in the third step, the two dimensions of the attentional style were inserted separately, and in the last step, the interaction terms between TP dimensions and internal and external attentional style were inserted separately. Finally, a simple slope analysis was performed to test the interaction effects. To highlight significant effects, 95% bootstrapped confidence intervals were calculated. If the confidence interval does not include zero, the effect is considered significant [52]. All *p* values of simple slope gradients are significant for *p* < 0.001.

## 5. Results

### 5.1. The Impact of Time Perspective on Social Network Addiction and the Moderating Role of Attentional Style

To investigate the hypothesis that attentional style moderates the relation between time perspective and social network addiction, we performed a series of hierarchical regression analyses using SPSS 22. Table 2 shows the results of the final moderation models.

The results of the hierarchical regression analyses (Table 3) show that the present-fatalistic (β = 0.16, *p* < 0.05), past-positive (β = 0.14, *p* < 0.05) and past-negative (β = 0.17, *p* < 0.05) orientations are positively associated with social network addiction, with an incremental variance of 11% (∆F = 5.66; *p* < 0.001). The internal attentional style is positively associated with social network addiction (β = 0.34, *p* < 0.001), while the external attentional style (β = −0.19, *p* < 0.01) is negatively associated with it. The introduction of the attentional style variable obtains an increase in the explained variance of 5% (∆F = 12.91, *p* < 0.001). We also found that the internal attentional style positively moderates the relationship between a past-positive perspective and social network addiction (β = 0.17, *p* < 0.05), while the external attentional style moderates it negatively (β = −0.20, *p* < 0.01). The moderation model explains 31% of the variance with an incremental variance of 8% (∆F = 2.05, *p* < 0.05). As shown in Figure 2 and Figure 3, high internal attentional style and high past-positive levels increase social network addiction, whereas high external attentional style and high past-positive levels decrease social network addiction.

### 5.2. The Impact of Time Perspective on Interpersonal Irritability and the Moderating Role of Attentional Style

Regarding time perspective, the present-fatalistic (β = 0.21, *p* < 0.001), and future (β = 0.21, *p* < 0.001) perspectives are positively associated with interpersonal irritability, with an incremental variance of 8% (∆F = 4.17; *p* < 0.001). Similar positive associations are found between interpersonal irritability and internal attentional style (β = 0.28, *p* < 0.001), while negative associations are seen with external attentional style (β = −0.21, *p* < 0.01). The introduction of attentional style variables obtains an increase in the explained variance of 7% (∆F = 8.76, *p* < 0.001). Regarding interaction terms, we found that internal attentional style positively moderates the relationship between the present-fatalistic orientation and interpersonal irritability (β = 0.16, *p* < 0.05), and conversely, external attentional style moderates it negatively (β = −0.27, *p* < 0.001). furthermore, the external attentional style positively moderates the future orientation and interpersonal irritability (β = 0.17, *p* < 0.05). The moderation model explains 23% of the variance with an incremental of 7% (∆F = 2.48, *p* < 0.01). Simple slope analyses (see Figure 4 and Figure 5) show that high internal attentional style and high levels of present-fatalistic thinking increase interpersonal irritability; conversely, high external attentional style and high levels of present-fatalistic thinking decrease it. Finally, (see Figure 6), high levels of external attentional style and high levels of future thinking decrease interpersonal irritability.

### 5.3. The Impact of Time Perspective on Elapsed Time and the Moderating Role of Attentional Style

The past-positive (β = 0.20, *p* < 0.01), and past-negative (β = 0.18, *p* < 0.05) orientations are positively associated with elapsed time, with an incremental variance equal to 8% (∆F = 4.17; *p* < 0.001). Furthermore, Internal attentional style is also positively associated with elapsed time (β = 0.283, *p* < 0.001). Introducing the attentional style variable, we obtain a 4% increase in the explained variance (∆F = 5.52, *p* < 0.01). Regarding the interaction terms, the external attentional style negatively moderates the relationship between the past-positive perspective and elapsed time (β = 0.19, *p* < 0.01); however, the moderation model explains only 19% of variance and does not show significant incremental percentages compared to Step 3 (∆F = 1.12, *p* = n.s.).

### 5.4. The Impact of Time Perspective on Social Performance Impairment and the Moderating Role of Attentional Style

The past-positive (β = 0.14, *p* < 0.05) perspective is positively associated with social performance impairment, while the future (β = −0.14, *p* < 0.05) perspective is negatively associated with it. The dimensions of time perspective obtain an increase in the variance equal to 8% (∆F = 4.24; *p* < 0.001). An internal attentional style is positively associated with social performance impairment (β = 0.38, *p* < 0.001) while the external attentional style shows a negative relation (β = −0.15, *p* < 0.05). The introduction of the attentional style variables shows an increase in the explained variance of 12% (∆F = 14.44, *p* < 0.001). Regarding the interaction terms, internal attentional style positively moderates the relationship between past-positive and social performance impairment (β = 0.21, *p* < 0.01), whereas external attentional style moderates it negatively (β = −0.18, *p* < 0.05). The moderation model explains 27% of the variance; however, also, in this case, it does not show significant incremental percentages concerning Step 3 (F = 1.63, *p* = n.s.). A similar trend has also emerged for the moderating effect of internal and external attentional style on the relationship between a past-positive orientation and Social performance impairment. As shown in Figure 7 and Figure 8, high levels of internal attentional style and high levels of past-positive thinking increase social performance impairment; conversely, a high level of external attentional style and high levels of past-positive thinking decrease it.

### 5.5. The Impact of Time Perspective on Social Network Anxiety and the Moderating Role of Attentional Style

In the last model, the role played by time perspective is analogous to the other models discussed above. Only a present-fatalistic (β = 0.19, *p* < 0.05) orientation is positively associated with social network anxiety, obtaining a 7% increase in variance (∆F = 3.77; *p* < 0.01). Internal attentional style is positively associated with Social Network Anxiety (β = 0.19, *p* < 0.05) while the external attentional style does not show a significant relation (β = −0.12, *p* = n.s.). An increase in the explained variance of 3% is observed (∆F = 3.61, *p* < 0.05). Regarding the interaction terms, the internal attentional style positively moderates the relationship between the present-fatalistic orientation and social network anxiety (β = 0.20, *p* < 0.05). The moderation model explains 23% of the variance, with a significant incremental percentage of 10% compared to Step 3 (∆F = 3.18, *p* < 0.001). Figure 9 shows that a high level of internal attentional style and high levels of present-fatalistic thinking increase levels of social network anxiety.

## 6. Discussion

### 6.1. Linear Associations among All the Variables of the Present Study

The results of the present study align with the previous literature [17,18]. Indeed, we found that the present-fatalistic and past-negative perspectives are positively associated with total score for social network addiction and with all its dimensions as measured by SNA-IS (i.e., interpersonal irritability; elapsed time; social performance impairment; and social network anxiety), and the future orientation was negatively associated with social performance impairment, thus corroborating our theoretical assumptions. Additionally, in our sample, people more oriented toward a fatalistic vision of their present and those more inclined to time orienting themselves toward negative aspects of their past are more addicted to social networking sites. More specifically, the involvement of people with high levels of present-fatalistic and past-negative thinking in using social networking sites is significantly related to impairment in all the core symptoms for social network addiction, such as salience, mood modification, time management, and social impairments. Thus, a possible explanation is coherent with the literature, which demonstrated that the present-fatalistic and past-negative perspectives are typical of individuals with low emotional stability, depression, and low self-esteem or external locus of control [23]. However, scholars also showed that those characteristics are also very particular to Internet-addicted individuals [53,54,55].

On the contrary, highly future-oriented people are less at risk for developing social network addiction, and this is might be because they are mostly emotionally stable or conscientious people, as suggested so far [23,56]. Unlike previous studies, we also found a significant linear association between past-positive orientation and particular aspects of social network addiction, i.e., the elapsed time. Specifically, we found that users who also score highly in this past-positive time dimension are also more inclined to spend more time on social networking sites. However, these results might be explained by the tendency of past-positive people to be proficient in time management [57], including the use of social networking sites in their daily activities rather than as a compulsive action.

No significant linear associations are observed in our data between a present-hedonistic orientation and social network addiction. However, coherently with our expectation, the results of the present study show that present-hedonistic-oriented people are not addicted to social networking sites, maybe because those kinds of applications might represent for highly present-hedonistic people more pleasure than noise [18,58].

Further results show that internal attentional style is negatively related to future, but this outcome might be due to the high prevalence in our sample of young people. Indeed, future-oriented young people might be deeply involved in their lifetime demands, focusing more on external than internal stimuli, helping them to gain academic achievement [59] and life satisfaction [60]. Conversely, internal attentional style correlated positively with social network addiction and all its negative impairments, and therefore it corroborates our hypothesis about the moderating role of attention in the relationship between time perspective and social network addiction.

### 6.2. The Moderating Role of Attentional Style in the Relation between Time Perspective and Social Network Addiction

The main goal of the present study was to investigate the moderating role of internal vs. external attentional style in the relationship between time perspective and social network addiction. The results of hierarchical regression analyses allow us to clarify these outcomes better. Indeed, we found that internal attentional style is a core variable in moderating the relationship between high levels of time perspective and a high level of social network addiction. This result is theoretically consistent with the cognitive definition of time perspective as a mental function that relies specifically upon the attentional process of selecting and orienting attention toward internal or external stimuli related to present or past events [23]. It also supports our hypothesis about the involvement of attention in regulating addictive behaviors toward Social Networking Sites. Unlike substance behavioral addictions, in which people focus on environmental cues [61] social network-addicted individuals are more oriented toward internal stimuli such as their intrusive thoughts or feelings. Therefore, the results are consistent with the literature about the definition of social network addiction as a psychopathological condition quite similar to obsessive compulsive disorder, depression, or anxiety [62]. Our outcomes might also reflect a situation of increased bottom-up attention and decreased top-down attention to internal cognitive and emotional events [63,64].

Moreover, the present study’s findings also gave a more detailed picture of the specific relations between different kinds of time orientations and the symptoms of social network addiction. Specifically, when high scores for internal attentional style are associated with high scores for a past-negative or past-positive orientation, the social network addiction total score and scores at the social performance impairment subscale increase. In addition, when high scores for internal attentional style are related to high scores for a present-fatalistic perspective, the scores for interpersonal irritability and social network anxiety subscale, enhance too. Thus, even if the involvement of past-negative in social network addiction corroborates previous literature outcomes [18], we conversely found that the past-positive and present-fatalistic orientations become crucial variables for social network addiction when considered in conjunction with attentional styles. Our results are understandable if we consider that the primary function of past-positive time orientation is to capitalize on a positive experience for present actions [23], and that a past-positive orientation is also related to time management strategies [57]. Thus, we might assume that people scoring highly in the Past-positive dimension, who are also more inclined to direct their attention toward intrusive thoughts, tend to compare their “instant” lives with those of other people [65], and so use social networking sites more intensively during their daily activities. However, the counterpart of this addictive tendency is the social impairment that leads people to neglect their family or work duties.

A high past-positive score could be the result of a cognitive evaluation between the present and past experiences. According to Zimbardo and Boyd [23], one of the characteristics of this factor is a nostalgic vision of the past. If the present experiences are less pleasant than past experiences, individuals may adopt some coping strategies to reduce the adverse effects of this evaluation. One of these strategies could be the tendency to dwell on the positive past. Another strategy could be to lock yourself in social networking site usage.

Similarly, when linked with present-fatalistic time orientation, internal attentional style leads people to more addictive behaviors toward social networking sites regarding interpersonal relationships or social anxiety. Indeed, scholars have demonstrated that the process of online social comparison might lead people to experience depression or other mood-related problems [66]. Finally, consistently with the prior literature [17,18,19], we also found that high levels of external attentional style and high levels of future thinking decrease interpersonal irritability. Furthermore, consistently with the tendency of future-oriented people to plan and gain their goals [32], we might deduce that future-oriented social networking site users are also more focused on processing external stimuli. Thus, they have fewer problematic behaviors in social networking site usage without cutting down the amount of time they spend on them or neglecting their social relationships.

## 7. Limitations

The present study has several limitations, and the results must be taken with caution. First, this is a cross-sectional research design, which cannot determine the cause–effect relationships between the variables. The choice of the roles of the variables of our moderation models was encouraged by the previous literature. Several pieces of research have already underlined the role of time perspective dimensions on various forms of social network addiction [17,18,19,34]. However, future research should be oriented towards longitudinal research designs that can give information on the directionality of the effects. A second limit is about our sample. This study was performed with a convenience and small sample consisting of university students, especially women, who are not representative of all populations of social networking site users. Although very interesting, it would be appropriate to test our results on a more representative and extensive sample. Finally, the data reported in this study come from self-report questionnaires. Several criticisms have been made of this method, especially in causing the common method bias [67]. Harman’s test was performed to reduce the chances of our data being affected by the standard method bias. The extracted factor explains 23% of the total variance or significantly below the threshold value of 50% [68].

## 8. Conclusions

This study aimed to understand the relationship between time perspective and social network addiction better. While taking the results into account with due care and having confirmed various previous knowledge, our contribution was also proposed to observe time perspective and attentional style interaction. The present study’s encouraging results allow us to better understand better psychological mechanisms underlying social network addiction, primarily focusing on the interaction between time perspective and attentional style. If confirmed by future research, these results could significantly assist clinicians, psychiatrists, psychologists, teachers, and all those who seek to combat such a problem in finding intervention strategies to reduce the harmful effects of social networking site abuse and promoting well-being [69]. As suggested by our results, in addition to being oriented towards the future, moving towards external attentional stimuli is the best strategy to prevent the pathological use of social networking sites. Our results, therefore, offer interesting insights and practical implications. Specific intervention programs, in subjects who show a high social network addiction, could increase and develop the concepts of anticipation, expectation, planning skills, indices of an orientation towards the future. Additionally, they could stimulate attentional processes in the external direction, and could represent not only a protective factor from social network addiction but at the same time constitute a valid therapeutic path.

## Figures and Tables

**Figure 1 jcm-10-03983-f001:**
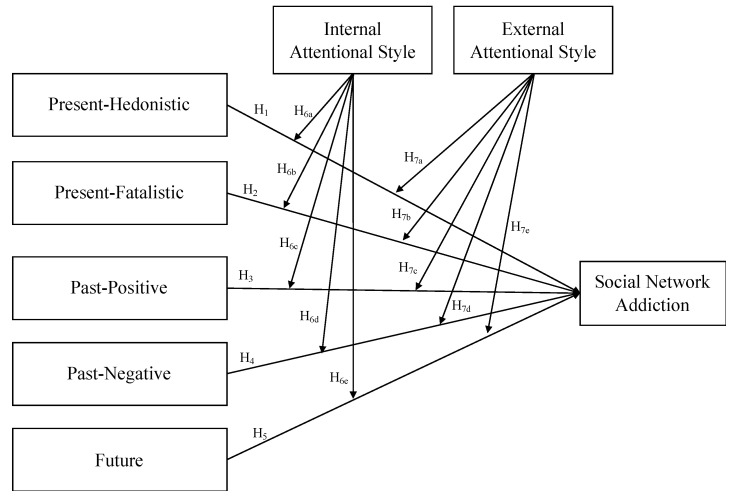
The hypothesized model.

**Figure 2 jcm-10-03983-f002:**
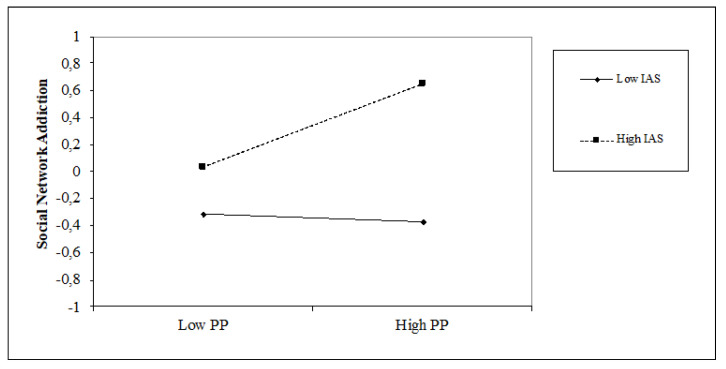
Simple slope of moderation effect of internal attentional style on the relationship between a past-positive orientation and social network addiction.

**Figure 3 jcm-10-03983-f003:**
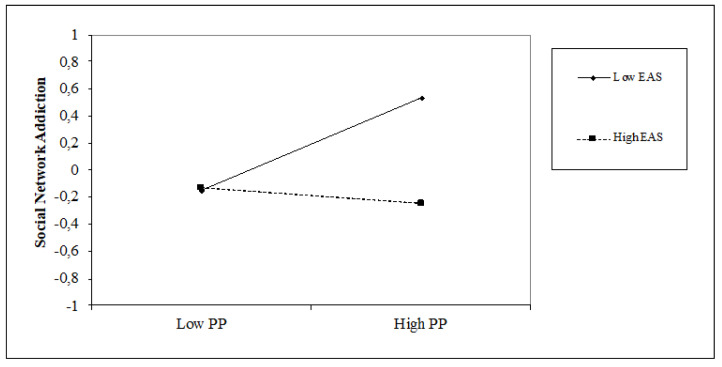
Simple slope of moderation effect of external attentional style on the relationship between a past-positive orientation and social network addiction.

**Figure 4 jcm-10-03983-f004:**
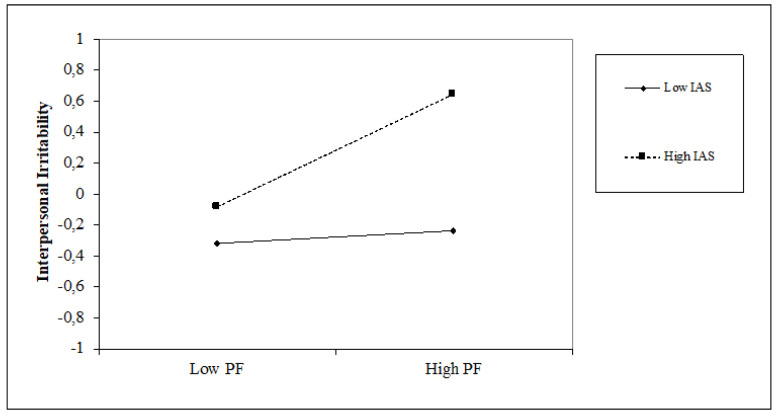
Simple slope of moderation effect of internal attentional style on the relationship between a present-fatalistic perspective and interpersonal irritability.

**Figure 5 jcm-10-03983-f005:**
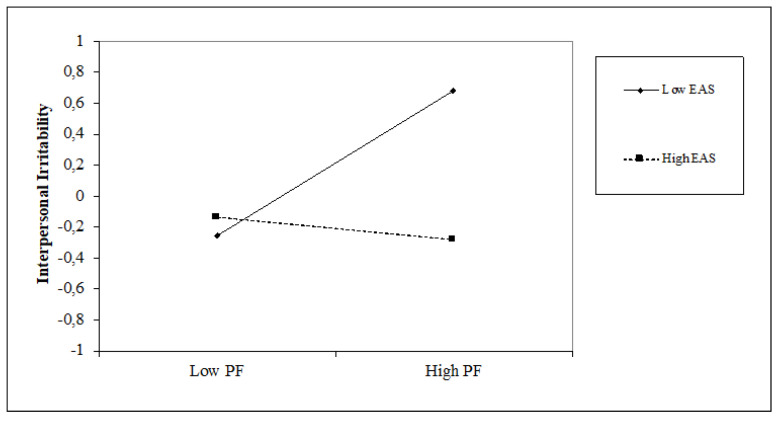
Simple slope of moderation effect of external attentional style on the relationship between a present-fatalistic perspective and interpersonal irritability.

**Figure 6 jcm-10-03983-f006:**
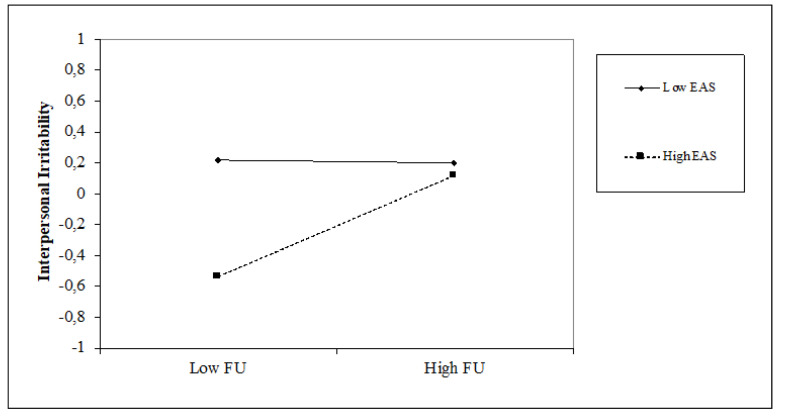
Simple slope of moderation effect of external attentional style on the relationship between a future perspective and interpersonal irritability.

**Figure 7 jcm-10-03983-f007:**
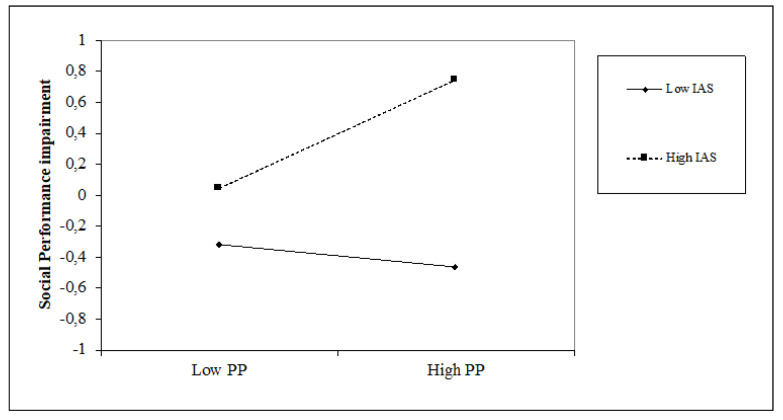
Simple slope of moderation effect of internal attentional style on the relationship between a past-positive orientation and social performance impairment.

**Figure 8 jcm-10-03983-f008:**
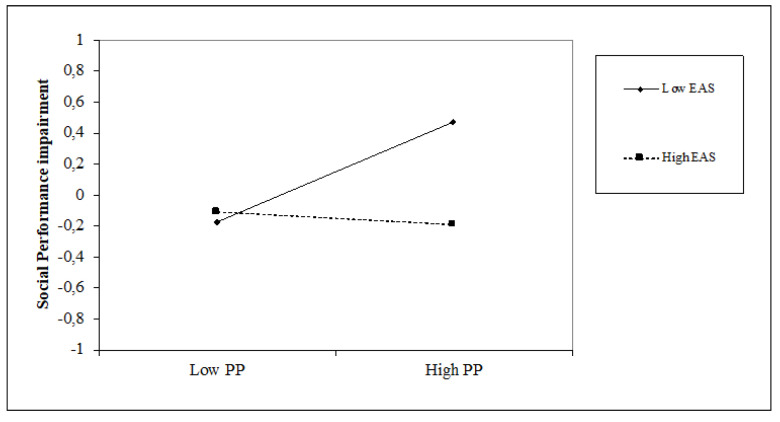
Simple slope of moderation effect of external attentional style on the relationship between a past-positive orientation and social performance impairment.

**Figure 9 jcm-10-03983-f009:**
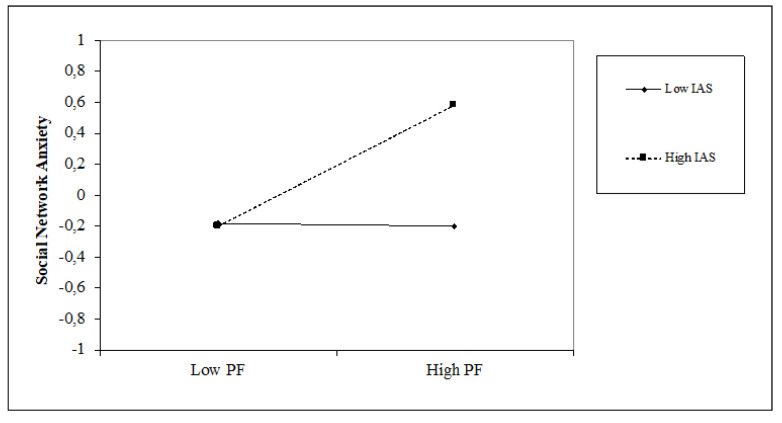
Simple slope of moderation effect of internal attentional style on the relationship between a present-fatalistic orientation and social network anxiety.

**Table 1 jcm-10-03983-t001:** Exploratory factor analysis of SNA-IS.

	Elapsed Time	Social Network Anxiety	Social Performance Impairment	Interpersonal Irritability
Item_07	0.77			
Item_08	0.63			
Item_06	0.51			
Item_05	0.32			
Item_13		0.82		
Item_14		0.60		
Item_16		0.59		
Item_15		0.44		
Item_11			−0.81	
Item_09			−0.78	
Item_10			−0.57	
Item_12			−0.31	
Item_04				−0.65
Item_01				−0.63
Item_02				−0.46
Item_03				−0.31
Explained variance	13.46	12.29	12.83	10.36

Note: *n* = 186; extraction method: principal axis factoring; rotation method: oblimin with Kaiser normalization; factor weights < 0.30 were omitted.

**Table 2 jcm-10-03983-t002:** Incremental changes between hierarchical regression steps.

	Social Network Addiction		Interpersonal Irritability		Elapsed Time		Social Performance Impairment		Social Network Anxiety	
Step	Adj R^2^	∆F	Adj R^2^	∆F	Adj R^2^	∆F	Adj R^2^	∆F	Adj R^2^	∆F
1	0.06	6.54 **	0.01	1.72	0.06	6.51 **	0.04	5.01 **	0.03	4.16 *
2	0.17	5.66 ***	0.09	4.17 ***	0.15	4.63 ***	0.12	4.24 ***	0.10	3.77 **
3	0.23	12.91 ***	0.16	8.76 ***	0.19	5.52 **	0.24	14.44 **	0.13	3.61 *
4	0.31	2.05 *	0.27	2.48 **	0.19	1.12	0.27	1.63	0.23	3.18 ***

Note: *n* = 186; * *p* < 0.05; ** *p* < 0.01; *** *p* < 0.001.

**Table 3 jcm-10-03983-t003:** Hierarchical regression analysis results.

	Social Network Addiction			Interpersonal Irritability			Elapsed Time			Social Performance Impairment			Social Network Anxiety		
	Beta	LL	UL	Beta	LL	UL	Beta	LL	UL	Beta	LL	UL	Beta	LL	UL
*Covariates*															
- Gender	0.23 ***	0.11	0.36	0.13	−0.00	0.27	0.21 **	0.07	0.35	0.21 **	0.08	0.35	0.18 *	0.04	0.32
- Age	−0.08	−0.21	0.05	0.01	−0.12	0.14	−0.13	−0.27	0.01	−0.06	−0.19	0.07	−0.04	−0.18	0.09
*Time perspective*															
- Present-Hedonistic (PH)	−0.02	−0.17	0.17	−0.06	−0.21	0.08	−0.012	−0.16	0.14	−0.03	−0.17	0.12	0.02	−0.13	0.17
- Present-Fatalistic (PF)	0.16 *	0.01	0.31	0.20 **	0.05	0.36	0.098	−0.06	0.26	0.04	−0.11	0.19	0.19 *	0.04	0.35
- Past-Positive (PP)	0.14 *	0.01	0.28	−0.01	−0.15	0.13	0.20 **	0.05	0.34	0.14 *	0.01	0.28	0.10	−0.04	0.24
- Past-Negative (PN)	0.17 *	0.02	0.31	0.13	−0.02	0.28	0.18 *	0.02	0.34	0.10	−0.05	0.25	0.11	−0.05	0.26
- Future (FU)	0.02	−0.11	0.16	0.16 *	0.02	0.30	−0.01	−0.15	0.14	−0.14 *	−0.28	−0.01	0.10	−0.04	0.24
*Attentional style (AS)*															
- Internal (I)	0.34 ***	0.19	0.49	0.28 ***	0.12	0.43	0.23 ***	0.07	0.39	0.39 ***	0.24	0.54	0.19 *	0.03	0.34
- External €	−0.19 **	−0.33	−0.05	−0.21 **	−0.35	−0.06	−0.12	−0.28	0.03	−0.15 *	−0.29	−0.01	−0.12	−0.27	0.02
*Interaction terms*															
- PH × IAS	−0.08	−0.22	0.07	−0.15	−0.29	0.01	−0.01	−0.16	0.15	−0.13	−0.27	0.03	0.011	−0.14	0.16
- PF × IAS	0.12	−0.03	0.26	0.16 *	0.01	0.31	0.05	−0.11	0.20	0.02	−0.13	0.17	0.20 *	0.05	0.35
- PP × IAS	0.17 *	0.03	0.29	0.056	−0.08	0.19	0.08	−0.07	0.21	0.21 **	0.07	0.33	0.22 **	0.07	0.34
- PN × IAS	0.06	−0.09	0.22	0.024	−0.13	0.18	0.03	−0.13	0.2	0.06	−0.09	0.22	0.10	−0.06	0.26
- FU × IAS	−0.10	−0.23	0.03	−0.06	−0.20	0.08	−0.08	−0.22	0.07	−0.07	−0.21	0.07	−0.12	−0.26	0.02
- PH × EAS	0.07	−0.09	0.22	0.20 *	0.02	0.35	−0.07	−0.24	0.11	0.07	−0.10	0.23	0.09	−0.09	0.25
- PF × EAS	−0.08	−0.23	0.07	−0.27 **	−0.40	−0.09	0.08	−0.08	0.24	−0.01	−0.16	0.14	−0.17 *	−0.32	−0.01
- PP × EAS	−0.20 **	−0.35	−0.06	−0.18 *	−0.33	−0.03	−0.19 **	−0.35	−0.04	−0.18 *	−0.33	−0.03	−0.06	−0.22	0.09
- PN × EAS	−0.11	−0.23	0.05	−0.03	−0.17	0.12	−0.12	−0.26	0.04	−0.13	−0.26	0.03	−0.04	−0.18	0.11
- FU × EAS	0.11	−0.03	0.23	0.17 *	0.02	0.29	0.06	−0.09	0.19	0.12	−0.02	0.25	0.01	−0.13	0.14
Adjusted R^2^	0.31			0.27			0.19			0.27			0.23		

Note: *n* = 186; * *p* < 0.05; ** *p* < 0.01; *** *p* < 0.001; LL = 95% bootstrap lower limit, UL = 95% bootstrap upper limit.

## Data Availability

The data presented in this study are available on request from the corresponding author.

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
