# Peer review of "Does Attentional Style Moderate the Relationship between Time Perspective and Social Network Addiction? A Cross-Sectional Study on a Sample of Social Networking Sites Users"

_jcm, 2021, doi:10.3390/jcm10173983_

Round 1

Reviewer 1 Report

This is a timely and relevant study with very interesting results. I have just some minor comment that I expose below:

In 2.2. Time Perspective and Attention. Please, define at the begining of the paragraph top-down or bottom-up attentional style.

In the SNA measure heading you said "This questionnaire offers both an SNA total score and four different scores related to specific symptoms, measured in four subscales (four items each): Interpersonal irritability (example of item: You try to cut down the amount of time you spend on SNSs and fail); Elapsed time (example of item: You find that you stay online on SNSs longer than you intended); Social Performance Impairment (example of item: You neglect household chores to spend more time on the SNSs); and Social Network Anxiety (example of item: You feel depressed, moody, or nervous when you are offline, which goes away once you are back on the SNSs)." Have you performed a Exploratory Factor Analysis to ensure that those 4 factors emerged in your study? I think that is very important to ensure that you are really reporting the four expected factors. And more taking into account that is your dependent variable and that is an adaptation of another scale.

Data analyses: I think that complementary relevant analyses to perform are moderation analyses with process for spss (you can use both, 1st or second model of SPSS). This is an analysis especially performed to corroborate moderation hypotheses, and it offers the possibility to generate plots for the interaction effects. Here you are a link that explain how to use it: https://www.youtube.com/watch?v=p0UbBJwFoeA with model 1 (one moderator- you will need in this case to run two different moderating analyses, one for each attentional style as moderator). Model 2 it the same but with the inclusion of two moderators in the same model (as your two attentional styles). 

Please, add as limitation the limited sample.

Please, provide some pratical implications of your interesting study and findings.

Figures are blurry and cannot be correctly seen. Please, improve them.

Good luck with your research!

Author Response

Reviewer 1:

1) In 2.2. Time Perspective and Attention. Please, define at the beginning of the paragraph top-down or bottom-up attentional style.

1) Thanks for this suggestion. We inserted at the beginning of the paragraph the definition of top-down and bottom-up attentional style.

2) In the SNA measure heading you said "This questionnaire offers both an SNA total score and four different scores related to specific symptoms, measured in four subscales (four items each): Interpersonal irritability (example of item: You try to cut down the amount of time you spend on SNSs and fail); Elapsed time (example of item: You find that you stay online on SNSs longer than you intended); Social Performance Impairment (example of item: You neglect household chores to spend more time on the SNSs); and Social Network Anxiety (example of item: You feel depressed, moody, or nervous when you are offline, which goes away once you are back on the SNSs)." Have you performed an Exploratory Factor Analysis to ensure that those 4 factors emerged in your study? I think that is very important to ensure that you are really reporting the four expected factors. And more taking into account that is your dependent variable and that is an adaptation of another scale.

2) Thank you for this suggestion. We performed and added an exploratory factor analysis. Results confirm a correlated four-factor structure.

3) Data analyses: I think that complementary relevant analyses to perform are moderation analyses with process for SPSS (you can use both, 1st or second model of SPSS). This is an analysis especially performed to corroborate moderation hypotheses, and it offers the possibility to generate plots for the interaction effects. Here you are a link that explain how to use it: https://www.youtube.com/watch?v=p0UbBJwFoeA with model 1 (one moderator- you will need in this case to run two different moderating analyses, one for each attentional style as moderator). Model 2 it the same but with the inclusion of two moderators in the same model (as your two attentional styles). 

3) Thanks for the suggestion. Unfortunately, PROCESS allows to test models have only one independent variable and one dependent variable. In our paper we want to test the five dimensions of the Time perspective simultaneously. However, we added bootstrap confidence intervals as suggested by Hayes (2018).

4) Please, add as limitation the limited sample.

4) Thanks for this suggestion. We added sample limits in Limitations section.

5) Please, provide some practical implications of your interesting study and findings.

5) Thanks for this suggestion. In the conclusion we added the practical implications

6) Figures are blurry and cannot be correctly seen. Please, improve them.

6) You're right. We improved the quality of all figures.

Reviewer 2 Report

This study examined the role of attentional style as a moderator variable between temporal perspective and social network addiction since little is known about users' cognitive variables involved in this kind of addictive behavior. In addition, this study suggested social network addicted users are more oriented toward internal stimuli. Generally, it was well written, but several weaknesses might have diminished the contribution of the current study.

The main point is that your study should focus on the moderation effects and consider deleting other unrelated parts.

Introduction

Line 37-38, "with an increasing percentage of 9% year to year," would be confusing. I would suggest changing 45% of the world population to the actual population number.

Too many abbreviations were introduced in the introduction part. I would suggest you spell out some of them (e.g., PF), Present-Hedonistic (PH), Past-Negative (PN), Past-Positive (PP), Future (FU), TP, and even SNA). They are pretty much unnecessary and cause trouble for readers. I have to go back and forth to see the definition for these abbreviations. See the article if you are interested.

https://www.psychologicalscience.org/observer/alienating-the-audience-how-abbreviations-hamper-scientific-communication

Method

One key question for this study is that you only have 186 samples while so many tests were done, which leads to suspicion of data dredging.

Some results presented in the texts were unrelated to the main study questions. For example, lines 292-294 talked about gender and age.

In addition, the correlation might not be beneficial when you have the regression results.

Author Response

Reviewer 2:

This study examined the role of attentional style as a moderator variable between temporal perspective and social network addiction since little is known about users' cognitive variables involved in this kind of addictive behavior. In addition, this study suggested social network addicted users are more oriented toward internal stimuli. Generally, it was well written, but several weaknesses might have diminished the contribution of the current study.

The main point is that your study should focus on the moderation effects and consider deleting other unrelated parts.

Introduction

1) Line 37-38, "with an increasing percentage of 9% year to year," would be confusing. I would suggest changing 45% of the world population to the actual population number.

1) You are right, we added actual population number

2) Too many abbreviations were introduced in the introduction part. I would suggest you spell out some of them (e.g., PF), Present-Hedonistic (PH), Past-Negative (PN), Past-Positive (PP), Future (FU), TP, and even SNA). They are pretty much unnecessary and cause trouble for readers. I have to go back and forth to see the definition for these abbreviations. See the article if you are interested.

https://www.psychologicalscience.org/observer/alienating-the-audience-how-abbreviations-hamper-scientific-communication

2) Thanks for your suggestions. We extended abbreviations.

Method

3) One key question for this study is that you only have 186 samples while so many tests were done, which leads to suspicion of data dredging.

In this study we have one simple with 186 respondents. Simple consists of volunteers and not financial compensation or university credits were expected. No data dredging was done. Data dredging usually involves the collection of such a large number of subjects that it can influence the p level. In psychological research, 186 subjects are considered a small sample. Simple limits were added in limitations section.

4) Some results presented in the texts were unrelated to the main study questions. For example, lines 292-294 talked about gender and age.

4) You're right. We removed all sections unrelated to the main study questions.

5) In addition, the correlation might not be beneficial when you have the regression results.

5) Thank you we removed correlation section

Round 2

Reviewer 2 Report

The authors have addressed my comments appropriately